# An Evaluation of Street Dynamic Vitality and Its Influential Factors Based on Multi-Source Big Data

**Xin Guo [1], Hongfei Chen [1,2,*] and Xiping Yang [1,2]** 

[1]   School of Geography and Tourism, Shaanxi Normal University, Xi'an 710119, China;
      guoxin001@snnu.edu.cn (X.G.); xpyang@snnu.edu.cn (X.Y.)
[2]   Shaanxi Key Laboratory of Tourism Informatics, Xi'an 710119, China
[*]   Correspondence: chenhongfei@snnu.edu.cn; Tel.: +86-13572858318

**Abstract:** Urban vitality is an important indicator of urban development capacity. Streets' metrics can depict intro-urban fabrics and physiognomy in detail, and thus street vitality affected by street metrics is a concrete manifestation of urban vitality. However, few studies have evaluated dynamic vitality or explored how it is influenced by land use. To bridge this gap, we fully evaluated street dynamic vitality and explored how to enhance the street dynamic vitality by changing the distribution and combination of land use. Specifically, we examined the street dynamic vitality and land use diversity in the main urban zone of Xining city in China using mobile communication and point of interest data, adopted optimized K-means clustering to identify street dynamic vitality types, evaluated the classification result based on vitality intensity and vitality stability, and explored the link between land use and dynamic vitality. Since vitality intensity limitations were found in describing street dynamic vitality, it was necessary to introduce vitality stability. We also found a positive correlation between the vitality intensity and land use density, there were outstanding traffic facilities in high-intensity vitality streets, and improving the abundance and uniformity of land use was beneficial to increase vitality stability. Overall, describing street vitality from a dynamic perspective can improve resource utilization efficiency and rationally plan layouts.

**Keywords:** street dynamic vitality; vitality intensity; vitality stability; land use; multi-source big data

## 1. Introduction

The world has ushered in the "urban era" where the proportion of urban population rising sharply to now exceed that of the rural population [1]. Joseph E. Stiglitz, the Nobel Laureate in Economics, stated that: "The high-tech development in the United States and China's urbanization will be two key factors affecting the process of human society development in the 21st century." The scale of Chinese urbanization is large, and its impact is far-reaching. Guided by people-oriented and harmonious living concepts, Chinese urbanization has transformed from scale development to connotative growth [2]. Urban vitality plays an important role in meeting people's demands for high-quality living, and promoting urbanization connotative development [3]. Jacobs first proposed "urban vitality" as the intensity of human activity in city space, which is characterized by people conducting abundant activities and the urban space providing a favorable environment for activities [4]. As the basic unit of public and living spaces [5], street metrics—including geometry, neighboring block connectivity, and vegetation—have proved effective in discriminating urban typologies in intro-urban structures and morphologies [6,7]. Hence, street characteristics are the microscopic embodiment of urban fabrics and physiognomy, and street vitality affected by street properties is a concrete manifestation of urban vitality. Meanwhile, street vitality is a manifestation of social vitality in urban vitality which is composed of economic vitality, social vitality, and cultural vitality [8]. According to the concept of urban vitality, street vitality is the concentration of people and activities in street space. However, in recent years, the functions of urban streets have changed. Specifically,

the increase in trafficability, decrease in sociality, and imbalance in people flow have led to a weakening of people's willingness to play activities. Therefore, the exploration of street vitality and its changes demands attention in the field of urban research.

Traditional quantitative studies generally obtain research data through questionnaires, field surveys, on-site interviews, and photography to evaluate vitality through comparisons, semantic analysis, and expert scoring [9–11]. However, traditional data are so static that quantitative studies cannot shed light on continuous spatio-temporal behavior. By measuring the average activity density and evaluating only vitality intensity, static vitality is affected by extreme values, which reduces the accuracy of vitality area recognition and confuses vitality areas with similar overall vitality but different patterns of change. Further, the static vitality overestimates or underestimates the vitality value in a certain period. This leads to incorrect urban planning decisions that result in redundancy and the insufficient allocation of public resources.

Volunteered geographic big data overcome the shortcomings of static vitality and show a strong potential for new exploration of a city by providing huge and successive spatio–temporal datasets [12]. Accordingly, the volunteered geographic datasets are increasingly applied to research on human movement patterns [13–15] and human–space interaction characteristics [16,17]. The new data environment has proven beneficial in characterizing urban or street vitality. For example, mobile phone data, characterized by wide spatial coverage, real-time data collection, and continuous user tracking, have become an ideal data source for studying urban vitality at the micro-level [18–21]. The number or density of specific facilities obtained by point of interest (POI) data is also used to quantify urban vitality [22]. Moreover, social network check-in data can display real-time urban vitality maps through kernel density analysis [23]. Benefiting from the low deviation of location-based service data, the real intensity of activities can be expressed to quantify the vitality [24]. Similarly, continuous and most recognizable Baidu heat maps can be applied to calculate the population [25]. Although most of the extant studies have adopted dynamic datasets, the vitality fell into static vitality due to the static measurement method of vitality and the inclination toward vitality spatial distribution and the factors of vitality spatial differentiation.

Based on the necessity of settling various defects in static vitality and the feasibility provided by the application of volunteered geographic big data, dynamic vitality came into being. Dynamic vitality helps explain the overall vitality intensity while recognizing the characteristics of vitality changes over time and thus provides the possibility to describe street vitality comprehensively. We believe that street vitality originates from continuous activities. Thus, we take a temporal series of activity density as a representation of street dynamic vitality and consider the street dynamic vitality as an ability to attract people and activities while maintaining the attraction at all times. Wu et al. [26] proposed the concept of "temporal vitality" by exploring the spatio–temporal distribution of vitality and influential factors based on social media check-in data in Shenzhen. By utilizing the temporal and spatial information contained in volunteered geographic big data, research on street vitality has started to focus on the temporal and spatial distribution of street vitality [27,28]. By contrast, few have evaluated temporal dynamic vitality through a comprehensive index system.

Within the literature, streets with a short length, high pedestrian density, high functional mixing, diverse building ages, and high connectivity represent vibrance [29], and street size, open spaces, building density, and compact structures affect street vitality [30]. The literature has comprehensively discussed multiple factors affecting street vitality, including the physical environment [31–34], employment rates [35], accessibility, and connectivity [36]. However, research on the impacts of specific land use indicators on dynamic vitality is limited.

More studies have examined urban vitality over street vitality, and although the spatial vitality methods are rather developed, the research on temporal vitality, its evaluation index, and the factors that impact its changing patterns is also rare. Thus, based on the connotation

of dynamic vitality, we took streets as the research object, used subscribers' spatio–temporal movement information—provided by mobile communication data—to quantify dynamic vitality, introduced the indicator "vitality stability" and evaluated street dynamic vitality together with "vitality intensity". We also analyzed the internal mechanism that affects street dynamic vitality in detail from a land use perspective using POI data.

The purpose of this research is to deepen the understanding of street vitality by evaluating dynamic vitality and to address the shortcomings of previous studies focusing on static vitality. Further, examining street dynamic vitality is beneficial to grasp the characteristics of urban structure and physiognomy and provides scientific guidance for urban planning and layout design.

## 2. Study Area and Dataset

### 2.1. Study Area

Our study area was the main urban area of Xining city, Qinghai Province, China (Figure 1). Located on the eastern edge of the Qinghai–Tibet Plateau, Xining city is the only modernized central city in the region with a population of over one million [37]. As the capital, Xining city plays an important role in leading and promoting the economic development of Qinghai Province and the regional development of the Hehuang Basin [38]. The main urban area is in the Xining Basin in the upper reaches of the Huangshui River. This river runs through Xining city from west to east and its tributaries—the Beichuan River and Nanchuan River—run through the north and south of Xining city. The main urban area is 26 km long from north to south and 32 km wide from east to west. It includes Chengzhong, Chengbei, Chengdong, and Chengnan Districts and has a total area of 346 km$^2$. At the end of 2017, the population of the main urban area was 1.3084 million, and 0.071% of the land area held 21% of the population of Qinghai Province.

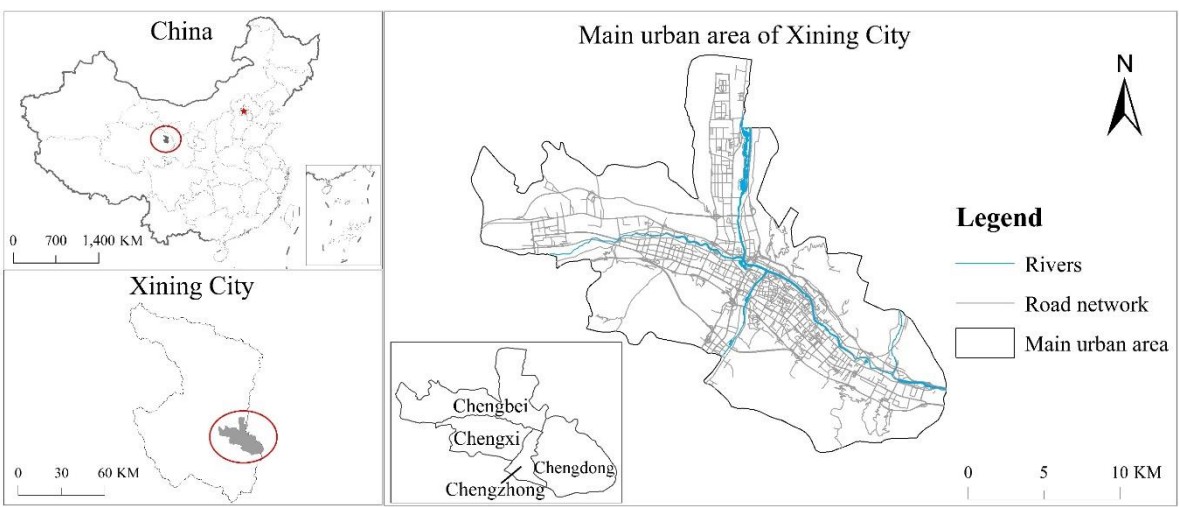

**Figure 1.** Study area location.

### 2.2. Dataset

We used the four datasets below to measure street dynamic vitality and the land use indices:

- The road network data. We obtained Xining city's road network data from the Open Street Map (OSM) website in 2018. The OSM data on major Chinese cities have a short renewal cycle and relatively high quality. Previous studies have used these data to measure the built environment [22] and represent urban forms and functions [39]. As there were too many details in the original road network data, we screened out the main roads from the messy road network (see Figure 2a) and took a buffer zone of 55 m on the both sides of the center as the street range [33].

- The mobile communication dataset. We obtained 258,814,289 communication records covering 24 hours a day from 566,163 mobile phone users in the main urban area. These were anonymized and were collected by one mobile phone operator in Xining city from 1 August 2018 to 6 August 2018. We only exploited the number of subscribers and did not include sensitive information. We extracted 4527 base stations in the area and illustrated their distribution in Figure 2b. The data contained when and where the users' mobile communication behavior occurred. We used the location of the base stations to approximately represent users' locations and to study street dynamic vitality.

- The urban spatial data. The Baidu Map [40] provides accurate and comprehensive urban spatial data for urban planning research [31], and we used the data to extract building names, spatial coordinates, floor space, height, basic outlines, and other information (see Figure 2c).

- The POI dataset. Since they are characterized by easy access, flexibility, and fine statistical granularity, the POI data reflect mixed land use and land development intensity better than traditional land use data. In recent years, POI data have mainly been used to measure the built environment [41]. We obtained POI data for cities in 2018 based on the AutoNavi Open Platform [42], the largest map navigation service provider in China, and selected 52,267 POIs at the street level (see Figure 2d). We grouped the POI data into 12 categories based on the POI category comparison table that was officially released on the AutoNavi Open Platform as well as urban land classification and construction land planning standards (GB50137-2011) [43]. The specific integration method was centered on the primary classification of the POI category comparison table, and the categories that were not included in the tertiary classification of the GB50137-2011 were merged with the corresponding categories. Table A1 in Appendix A shows the classification result of POIs in the main urban area.

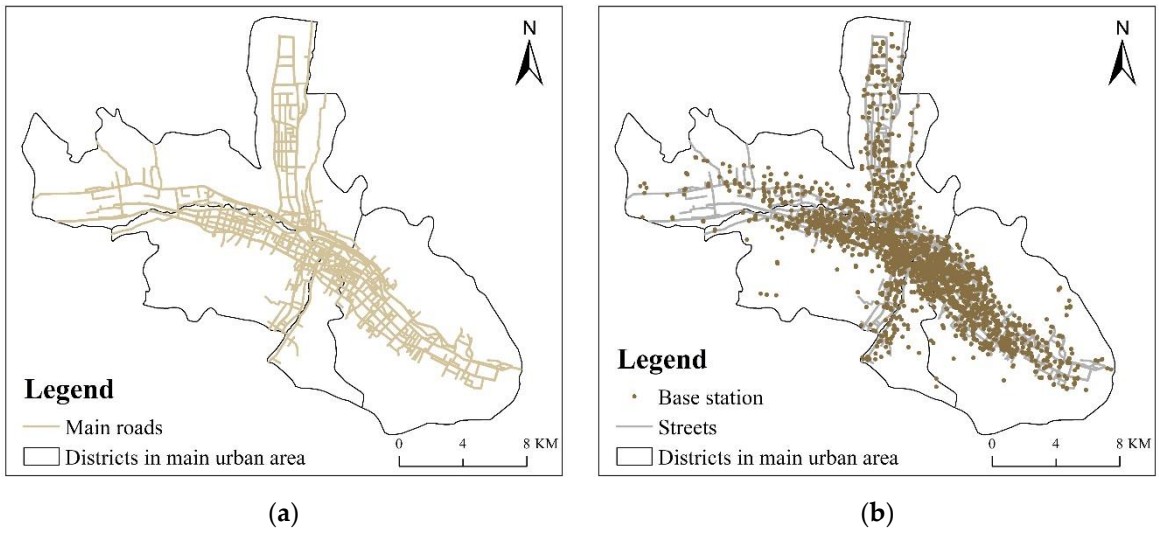

**(a)** **(b)**

**Figure 2.** *Cont.*

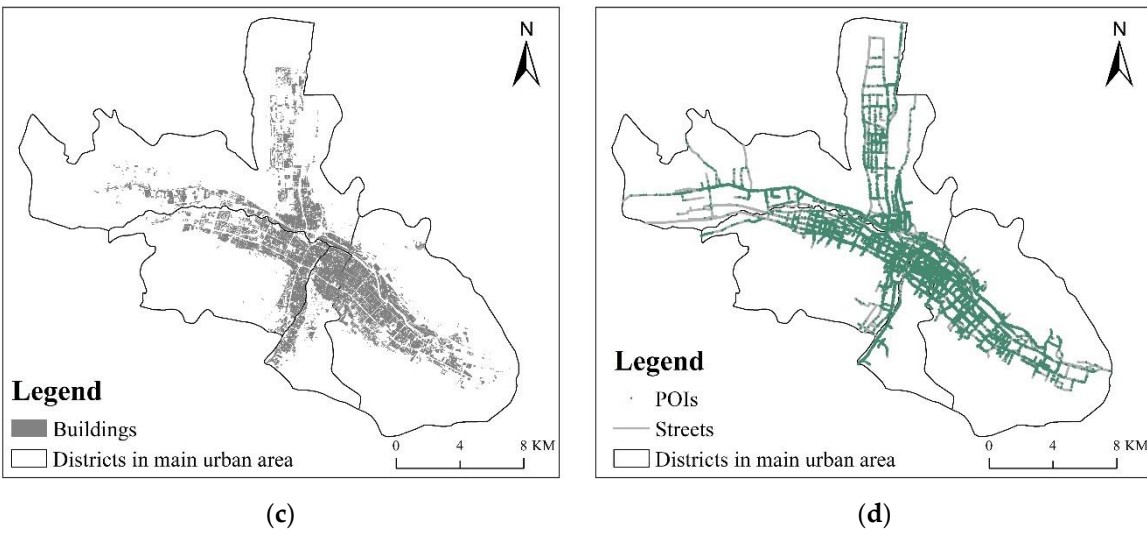

**Figure 2.** Spatial distribution of the elements from the four data sources. (**a**) Main roads; (**b**) Base station; (**c**) Buildings; (**d**) points of interest (POIs).

## 3. Methodology

### 3.1. Street Dynamic Vitality

We used the population density at each time period to represent street dynamic vitality. The urban spatial data reflected the three-dimensional space where vitality occurred, the Thiessen polygon approximately represented the service range of the base stations, and the mobile phone communication data at different time periods quantified street dynamic vitality.

The built environment affects the subjective feelings and activities of crowds [24]. The three-dimensional space enclosed by a road intersection is a complete space through which the elements within the built environment affect people's feelings and the integrity of their activities. Therefore, using the processed road network data, we considered the space enclosed by two road intersections as a street (streets with a length less than 200 m were not divided by intersections). Finally, we obtained 652 streets in the main urban area of Xining city.

The time units of the mobile communication data were accurate to the second, and the spatial sampling accuracy of the mobile communication was about 160 m in our study. Table 1 presents the information of the acquired mobile communication data.

**Table 1.** Mobile communication data for Xining City.

| Fields | Descriptions |
|---|---|
| ID | The unique and encrypted identification numbers of mobile phone subscribers |
| Start Time End Time | The time stamp when mobile communication occurred, accurate to the second |
| Longitude Latitude | The base station's location, where the subscribers were when the mobile communication occurred |

Data pre-processing involved deleting data with missing timestamps, removing personal trajectory information outside the scope of this research area, and eliminating the ping-pong phenomenon in neighboring base stations. It also included dividing the data into 12 periods at two-hour intervals and counting the population served by the base station by these 12 groups.

We allocated the population on the street equally following the spatial area in the Thiessen polygon [44]. As shown in Formula (1), the spatial area refers to the sum of the

block area and building area, excluding the ground floor of the building. The population in the smallest unit of the street is determined by the ratio of the spatial area of the unit to the total spatial area of the Thiessen polygon, and formula (2) is proposed. (The smallest unit of the street is the smallest irregular spatial unit formed by the intersection of the street and Thiessen polygon). Hence, street vitality can be expressed as population density per spatial area, so the vitality value at each time period is shown in Formula (3).

$$S_k = S_d + S_j(f-1), \tag{1}$$

$$X_i = Q \times S_{ik}/S_{bk}, \tag{2}$$

Formula (1) calculates the spatial area ($S_k$ is the dependent variable), where $S_d$ represents the block area, $S_j$ is the building area, and $f$ is the number of building floors. Formula (2) measures the smallest unit's allocated population ($X_i$ where $Q$ is the number of users served by the base station connected to the smallest unit, $S_{ik}$ is smallest unit's total spatial area, and $S_{bk}$ is the Thiessen polygon's spatial area corresponding to the smallest unit.

$$V = X/S, \tag{3}$$

Formula (3) measures the value of the street's vitality ($V$), where $X$ represents the number of subscribers on the street and S reflects the street's total spatial area.

### 3.2. K-means Clustering

Regular spatio–temporal patterns of crowd activity determine the continuity and dynamics of street vitality [26]. We focused on the changing patterns of street vitality and used the K-means clustering method to cluster streets with similar patterns of vitality. We used a distance-based non-hierarchical clustering algorithm that took the error function as a reference and divided the data into K sets, where distance measured the similarity of the time series datasets. The closer the distance between two objects, the greater is the similarity [45].

First, we selected K objects as the clustering center. Second, we calculated the distance of other sample objects from the K clustering center, and then divided the objects into the nearest clustering center to form K clusters. Third, we recalculated the mass center of each cluster. Lastly, we repeated steps two and three until the mass position center no longer changed or until we reached the set number of iterations. Formula (4) calculates the distance between two objects:

$$Dist(X,Y) = \sqrt{\sum_{j=1}^{n}(X_j - Y_j)^2}, \tag{4}$$

where $X$ and $Y$ represent the objects, and $j$ represents the dimension of the calculated object.

To optimize this method, the data need to be pre-processed with Z-score standardization to improve the clustering accuracy and the convergence speed of the min-imum error function. The elbow rule determined the value of "K", and the inflection point position was the optimal classification number "K". To ensure the stability of the final results and the maximum difference between the cluster classes, we used the maximum distance method to select the initial clusters.

### 3.3. Index System

- Street Dynamic Vitality Evaluation Index

Formula (5) shows an effective index to measure overall vitality intensity [27]; however, it cannot describe fluctuation characteristics of vitality. We therefore defined vitality stability as the ability of activities to maintain stability over time, and its quantitative

method is shown in Formula (6). We adopted vitality intensity and vitality stability to construct an index system for comprehensively evaluating street dynamic vitality.

$$ITS = \frac{\sum_{i=1}^{n} V_i}{n} \tag{5}$$

$$STB = \frac{ITS}{\left(\sum_{i=1}^{n}(V_i - ITS)^2/(n-1)\right)^{\frac{1}{2}}}, \tag{6}$$

where $V_i$ is the vitality value in each time period calculated using the same calculation as V in Formula (3), and n is the number of periods.

- Land Use Index

The diversity of street functions plays an important role in ensuring street vitality [4] (p. 150), and the diversity of land use determines the diversity of street functions to some extent. The term "diversity" was originally used to describe species diversity in ecology and was later used to describe land use diversity [46]. We employed a series of indicators that measured species diversity to construct a land use diversity evaluation index that includes density, richness, the Simpson index [47], and the main land use types. Density reflects the intensity of land development, where size depends on the number of POIs (Formula (7)); richness directly represents the number of land use types (Formula (8)); the Simpson index as a reference of uniformity for land use distribution (Formula (9)) [41]. And we define the main land use types as those with high proportions and important functions, and the selection method is shown in Formula (10).

$$Den = N/S, \tag{7}$$

$$Ric = (m-1)/\ln N, \tag{8}$$

$$Sim = 1 - \sum_{i=1}^{m} P_i^2, \tag{9}$$

$$Main = type(abs(sum\_topi(\{P1, P2, \cdots, Pi, \cdots, Pn\}) \geq \alpha)), \tag{10}$$

where $N$ represents the number of POIs, $S$ is a street's spatial area, $m$ represents the total number of land use types, $Pi$ is the proportion of each land use type, the $sum\_topi()$ function calculates the cumulative percentage, $\alpha$ is the cumulative percentage threshold, the $abs()$ function selects the value that meets the requirements, and the $type()$ function identifies the object corresponding to the value.

### 3.4. Methodology Framework

The activities change dynamically over time and occur in three-dimensional space. We chose the spatio–temporal dataset of the human movement provided by the mobile communication data and the floor information obtained by urban spatial data to measure street dynamic vitality. Then, we adopted the optimized K-means to classify the street vitality types according to the similarity of their dynamic vitality and evaluated the classification results by the dynamic vitality evaluation system formed by the vitality intensity and vitality stability. Finally, using POI data, we constructed a land use index system constituted by the density, richness, and Simpson index of land use, as well as the main land use types in preparation for discussing the impact of land use on street dynamic vitality types (Figure 3).

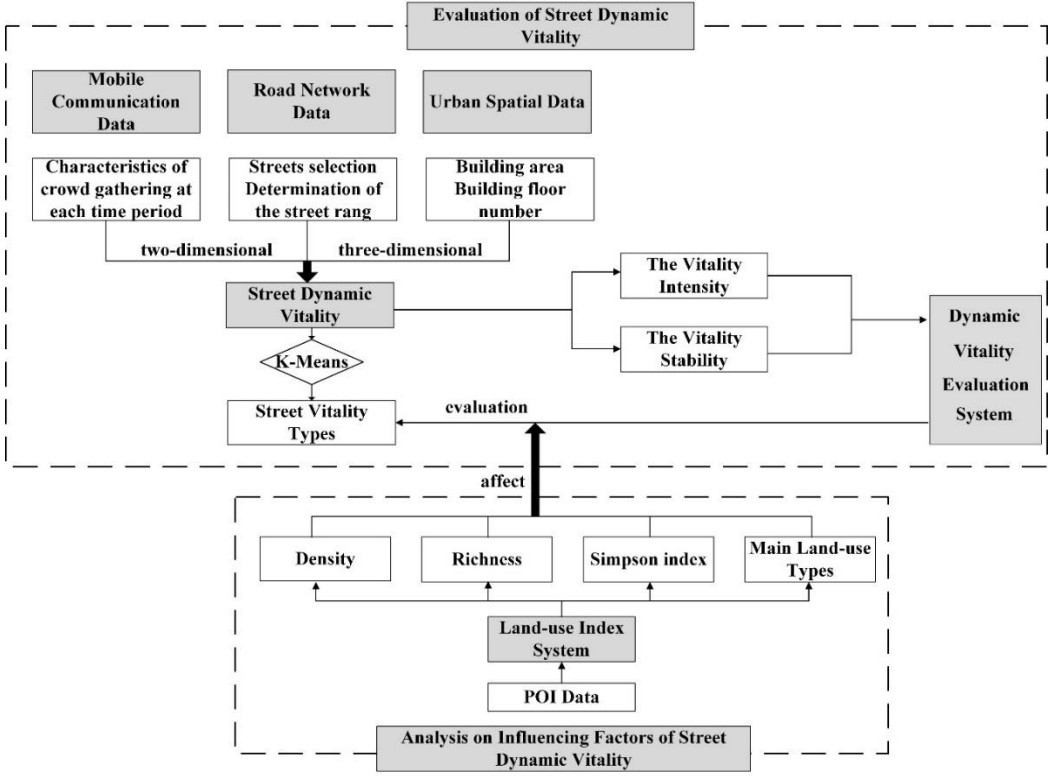

**Figure 3.** The methodology framework.

## 4. Results

### 4.1. Characteristics of Street Temporal and Spatial Vitality

Figure 4 shows that the variation in street vitality is in line with the rhythm of human daily lives. The vitality value decreases significantly during sleep times, and rises sharply during the morning rush hour. It peaks around noon and dip in the afternoon. When the evening rush hours or having entertainment at night, the vitality value rises and approaches the peak. Around 24:00, people stop outdoor activities, and street vitality decreases. Spatially, there are different distribution characteristics of the streets with high-value vitality in four districts of the main urban area. The high-value vitality streets are mainly concentrated in the shopping malls and commercial pedestrian areas in the Chengxi District, Beichuan River in the Chengbei District, parks and squares, shopping malls and Nanchuan River in the Chengzhong District, and shopping malls and schools in the Chengdong District. Therefore, a good consumption environment is conducive to the formation of street vitality in the main urban area.

Figure 5 indicates that from 0:00 to 6:00, the street vitality value in the city center is generally lower on weekdays than that on weekends. Except for the streets near the industrial parks in the Chengbei District and Chengdong District, the street vitality values in the peripheral area of the main urban area are almost the same as those on weekends. From 6:00 to 8:00, owing to the morning rush hour, the street vitality on weekdays is generally higher than that on weekends. During working hours, there are two-thirds of the streets with higher vitality on weekdays than on weekends, half of which with much higher vitality on weekdays than on weekends. After 18:00, the street vitality values on weekends generally exceed those on weekdays, which points to the fact that the street vitality at night mainly comes from the activities on weekends.

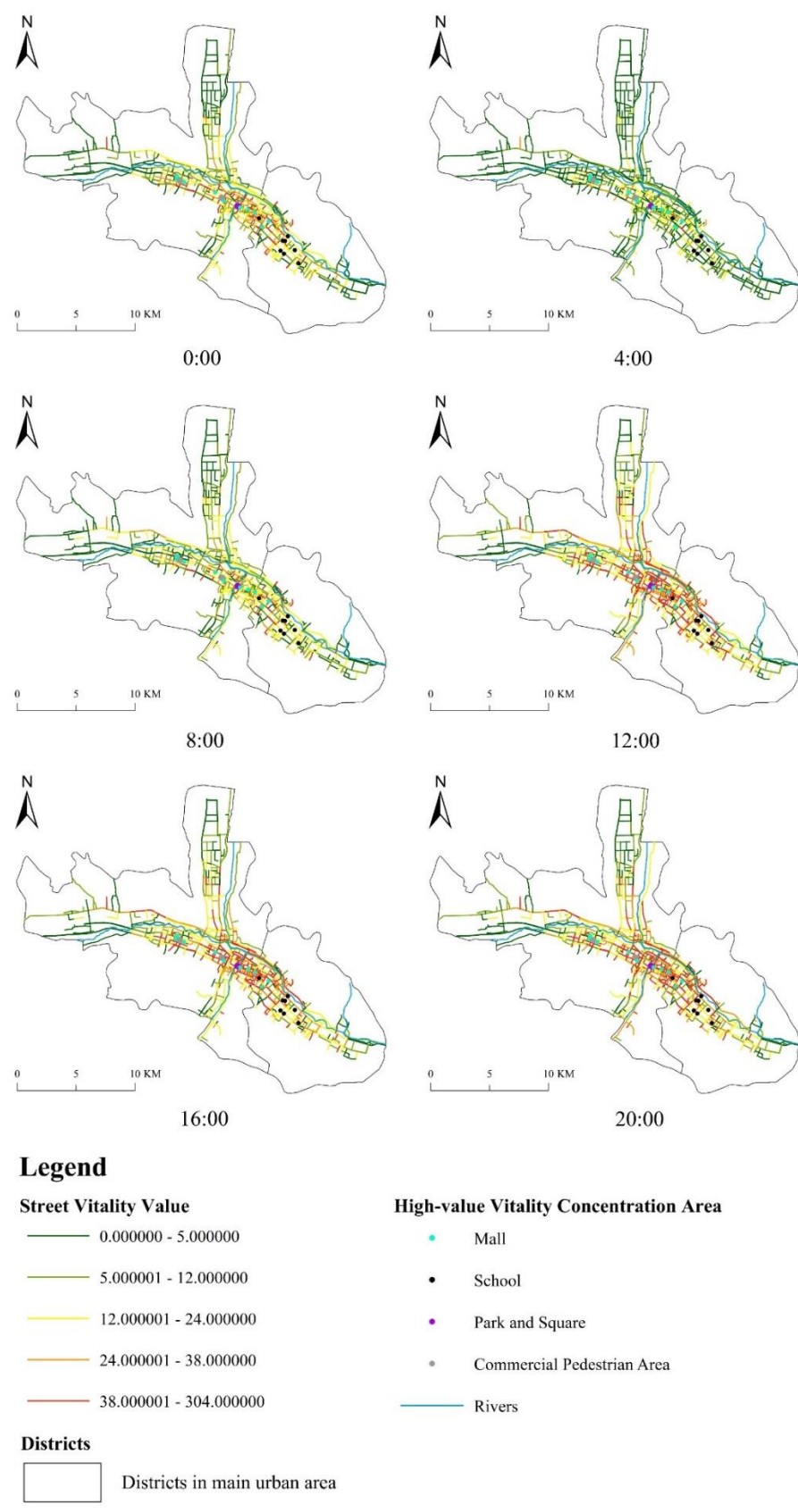

**Figure 4.** Spatio–temporal vitality of streets.

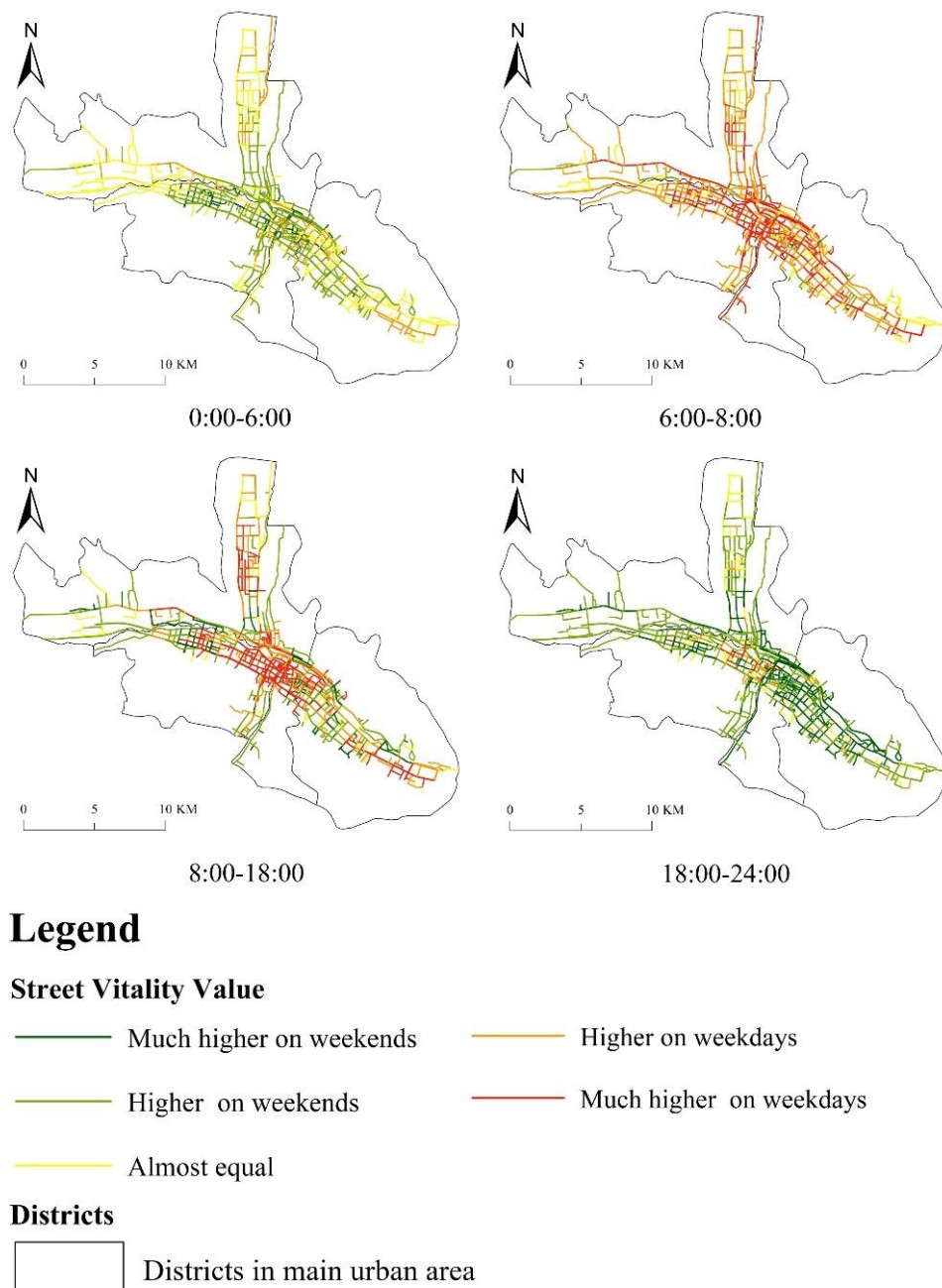

**Figure 5.** Comparison of street vitality values on weekdays and weekends.

*4.2. Street Vitality Types*

For the street vitality types, classified by K-means clustering, we used the elbow rule to determine the optimal number of street types. Figure 6 indicates that the street vitality types on weekdays and weekends are divided into four categories.

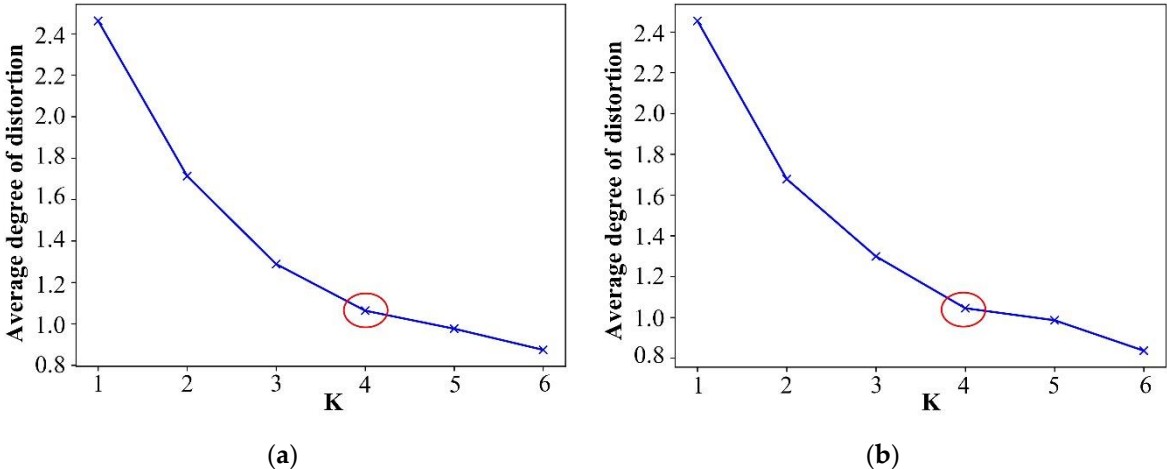

**Figure 6.** Best "K" value by the elbow rule. (**a**) Ideal "K" value on weekdays; (**b**) Ideal "K" value on weekends.

As seen in Table 2 where the evaluation index was calculated by Formulas (5) and (6), we define the street vitality types whose vitality intensity is higher than the average value of vitality intensity as high-intensity, and those below the average value as low-intensity. In the same vein, the classification of high-stability and low-stability depends on the average value of vitality stability. Following the above division rule, these categories were high-stability and high-intensity vitality streets (HSHI), high-stability and low-intensity vitality streets (HSLI), low-stability and high-intensity vitality streets (LSHI), and low-stability and low-intensity vitality streets (LSLI). The vitality stability of streets with similar vitality intensity was quite different. On weekdays, streets with stable vitality changes account for 79.8% of high-intensity vitality streets, while 20.2% of street vitality changes drastically. Further, 35.6% of low-intensity vitality streets' vitality changes slightly, and the remaining streets' vitality changes drastically. During weekends, streets with similar vitality intensity also show significant differences in stability. Thus, it is necessary to combine vitality intensity and stability to describe street dynamic vitality.

**Table 2.** Classification and evaluation of street vitality types.

| Street Vitality Types | | Number of Streets | Evaluation Index | |
|---|---|---|---|---|
| | | | ITS (Vitality Intensity) | STB (Vitality Stability) |
| | HSHI | 67 | 62.6536 | 2.2109 |
| Weekdays | HSLI | 202 | 29.0445 | 2.1290 |
| | LSLI | 366 | 7.9697 | 1.9410 |
| | LSHI | 17 | 132.4735 | 1.9489 |
| | HSHI | 67 | 63.0059 | 2.2836 |
| Weekends | HSLI | 189 | 29.2996 | 2.2090 |
| | LSLI | 382 | 8.2765 | 2.0092 |
| | LSHI | 14 | 132.3198 | 2.0547 |

*4.3. Temporal and Spatial Characteristics of Street Vitality Types*

As shown in Figure 7, the streets with high-intensity vitality occupy the prosperous areas in the city center, the vitality intensity decreases from the city center to the periphery, and the high and low vitality stability distribution are nested. HSHI and LSHI are distributed around the city center, where activity is dense at all times, and the HSHI is dominant. LSLI is widely distributed in the fringe area, and HSLI is concentrated around the Huangshui and Nanchuan Rivers.

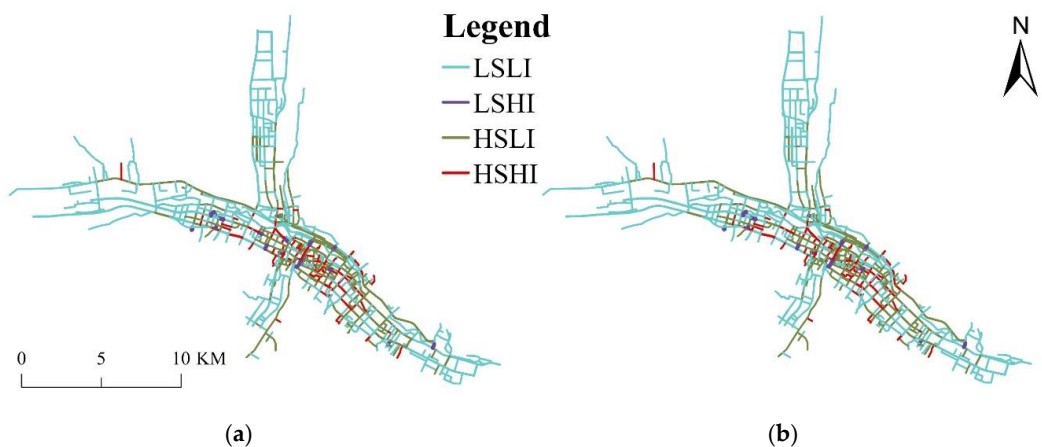

**Figure 7.** Spatial distribution of street vitality types. (**a**) Weekdays; (**b**) Weekends. "LSLI" represents streets with low-stability and low-intensity vitality; "LSHI" is an abbreviation for streets with low-stability and high-intensity vitality; "HSLI" describes the streets with high-stability and low-intensity vitality, and "HSHI" symbolizes the streets with high-stability and high-intensity vitality.

Figure 8 indicates that most street vitality types do not change from weekdays to weekends. Of the 30 streets with varying vitality types, including from HSLI to LSLI, they are mainly distributed in residential areas, where people's activities are irregular on weekends. The vitality stability of streets from LSLI to HSLI and LSHI to HSHI are improved, and these appear near the prosperous traffic arteries. For the streets from HSLI to HSHI and HSHI to HSLI, the vitality intensity changes significantly. Among these, there are prominent ornamental and entertainment facilities, and because of the significantly lower number of visitors on weekends, the vitality intensity of many schools and hospitals reduces during these time.

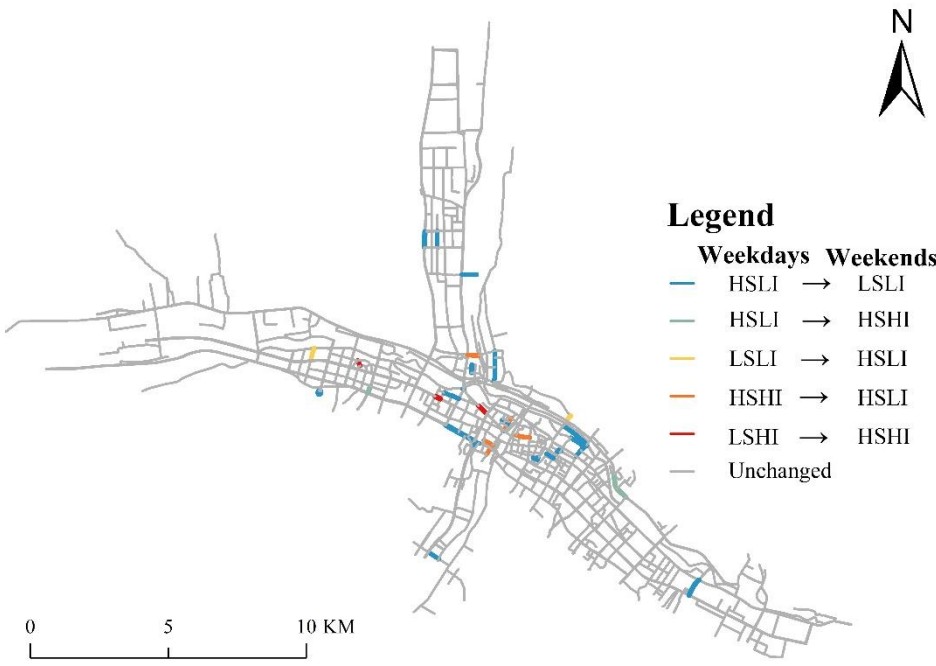

**Figure 8.** Spatial distribution of changes in street vitality types from weekdays to weekends.

Figure 9 shows that LSHI's peak first appears around 12:00, that the second peak emerges later on weekends than on weekdays. The daytime and night-time vitality values of LSHI streets are higher than those of the other types; yet, the vitality fluctuation is

large and stability is poor over a day. The vitality intensity of HSHI is slightly lower than that of LSHI; its vitality peak is not evident and the temporal distribution of the crowd is relatively even. HSLI and LSLI show lower vitality values, as they are far less likely to attract crowds than the first two types of streets, especially LSLI's extremely poor vitality status (its highest vitality intensity value is less than 50 people per hectare). Figure 10 compares the evaluation indicators of street vitality types on weekdays and weekends. From weekdays to weekends, the vitality intensity changes less perceptively, but the vitality stability changes significantly. Moreover, the stability of street vitality types is higher on weekends than on weekdays, which indirectly reflects that vitality performance is more reasonable on weekends than on weekdays, which effectively reduces street pressure.

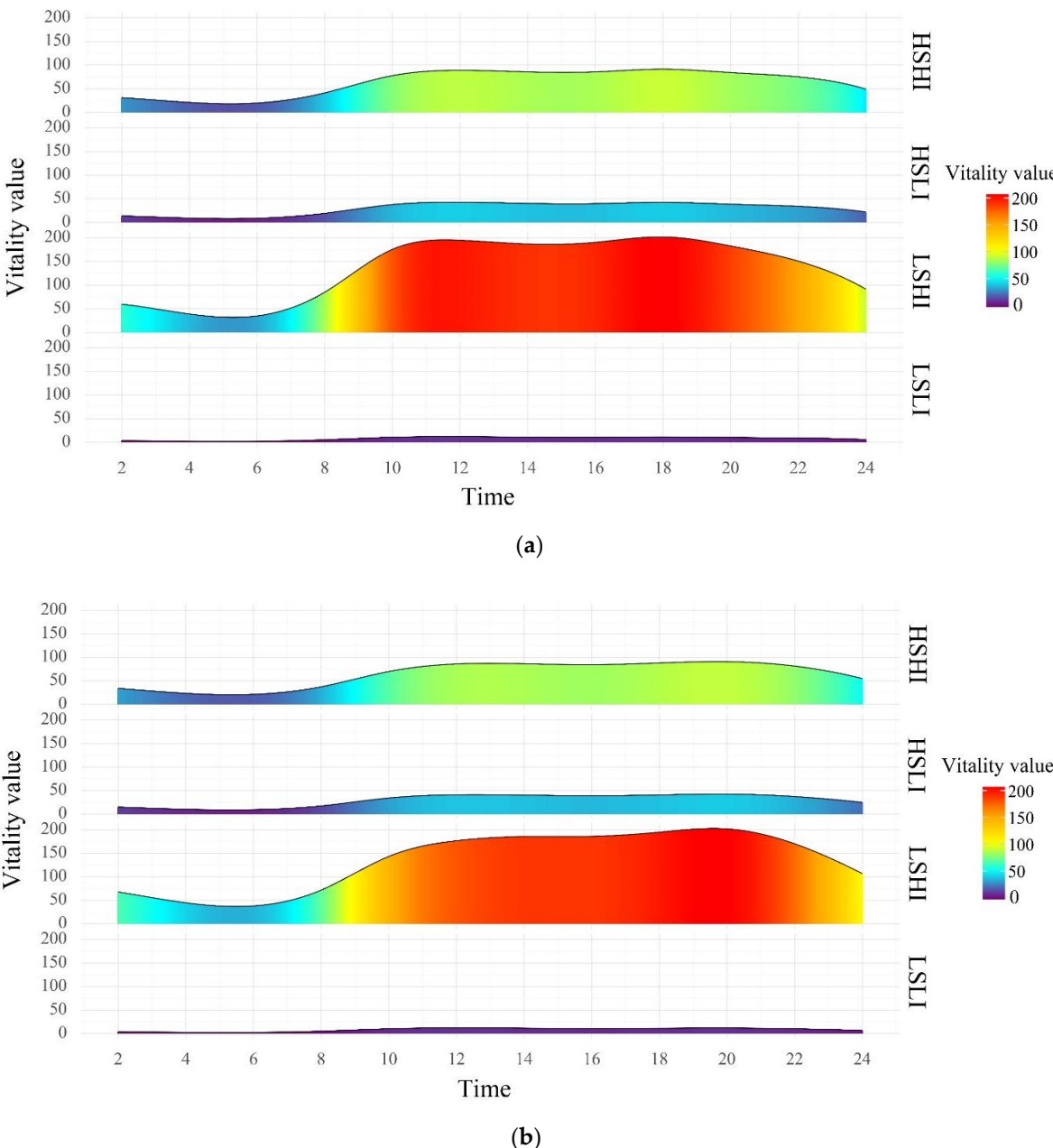

**Figure 9.** Temporal characteristics of street vitality types. (**a**) Weekdays; (**b**) Weekends.

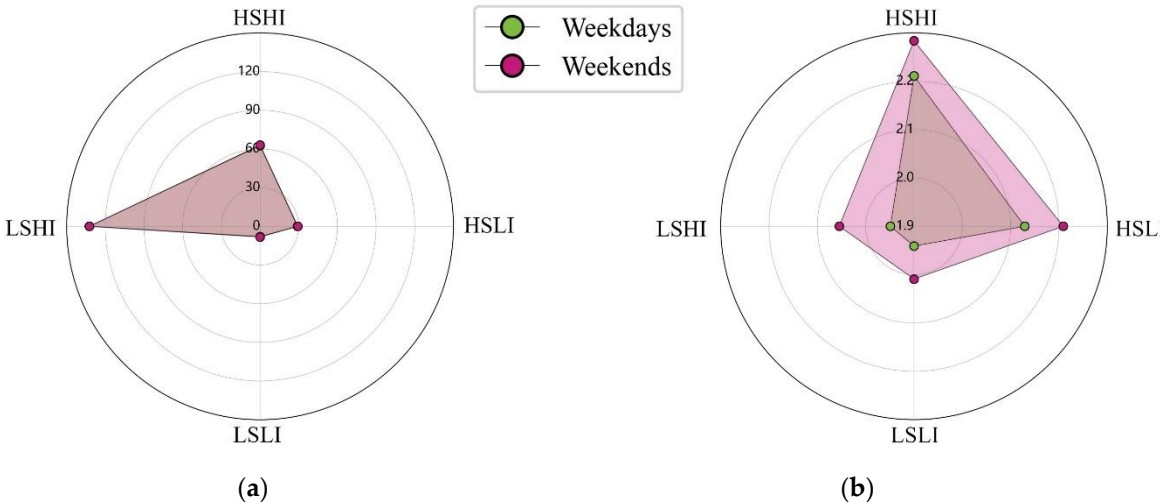

**Figure 10.** Comparison of the dynamic evaluation index of street vitality types on weekdays and weekends. (**a**) Vitality intensity; (**b**) Vitality stability.

### 4.4. Influential Factors of Street Vitality Types

We used the POI indicators to analyze the influencing factors of land use on street dynamic vitality. Table 3 shows the results of Xining City's land use indicators introduced in Section 3.3. As five streets had missing POI data, we calculated the average values of their vitality type indicator. Figure 11 shows the land use composition of the street vitality types in Xining city's main urban area.

**Table 3.** Land use characteristics of street types.

| Category | | Land Use Index | | | | |
|---|---|---|---|---|---|---|
| | | Count | Density | Richness | Simpson | Main Type |
| Weekdays | HSHI | 7149 | 5.7797 | 2.0470 | 0.7813 | SR, LS, CS, TS |
| | HSLI | 25,032 | 5.7589 | 1.8895 | 0.7566 | SR, LS, CS, RA |
| | LSHI | 1392 | 8.6611 | 1.8496 | 0.7025 | SR, LS, CS |
| | LSLI | 18,694 | 3.0913 | 1.5528 | 0.6342 | SR, LS, CS, RA |
| Weekends | HSHI | 7188 | 5.8956 | 1.9898 | 0.7640 | SR, LS, CS, TS |
| | HSLI | 23,565 | 5.8388 | 1.9148 | 0.7602 | SR, LS, CS, RA |
| | LSHI | 1240 | 9.1098 | 1.8922 | 0.7317 | SR, LS, CS |
| | LSLI | 20,274 | 3.1513 | 1.5627 | 0.6393 | SR, LS, CS, RA |

Note: Shopping service (SR); Life service (LS); Catering service (CS); Residential area (RA); Traffic service (TS).

Table 3 shows the land use indicators of different vitality types of streets in the main urban area of Xining city. The density calculated by the number of POIs reflects the intensity of land development, the richness measures the diversity of street function, and the Simpson index describes the uniformity of the street function distribution. The main land use types are the key to attracting vitality and dominate the street functions. Combining Tables 2 and 3, we can see that POI density is positively correlated with the vitality intensity of the street vitality types: the POI density of the high-vitality streets is higher than that of the low-vitality streets. In short, the intensity of land development positively affects street vitality intensity. Vitality stability is positively correlated with POI richness and the Simpson index. The richness and Simpson index of the high-stability streets are higher than those of the low-stability streets. Thus, the richer and more uniform the land use, the more it can meet the activity needs of different groups of people at different times, and the higher is the sustainability of street vitality.

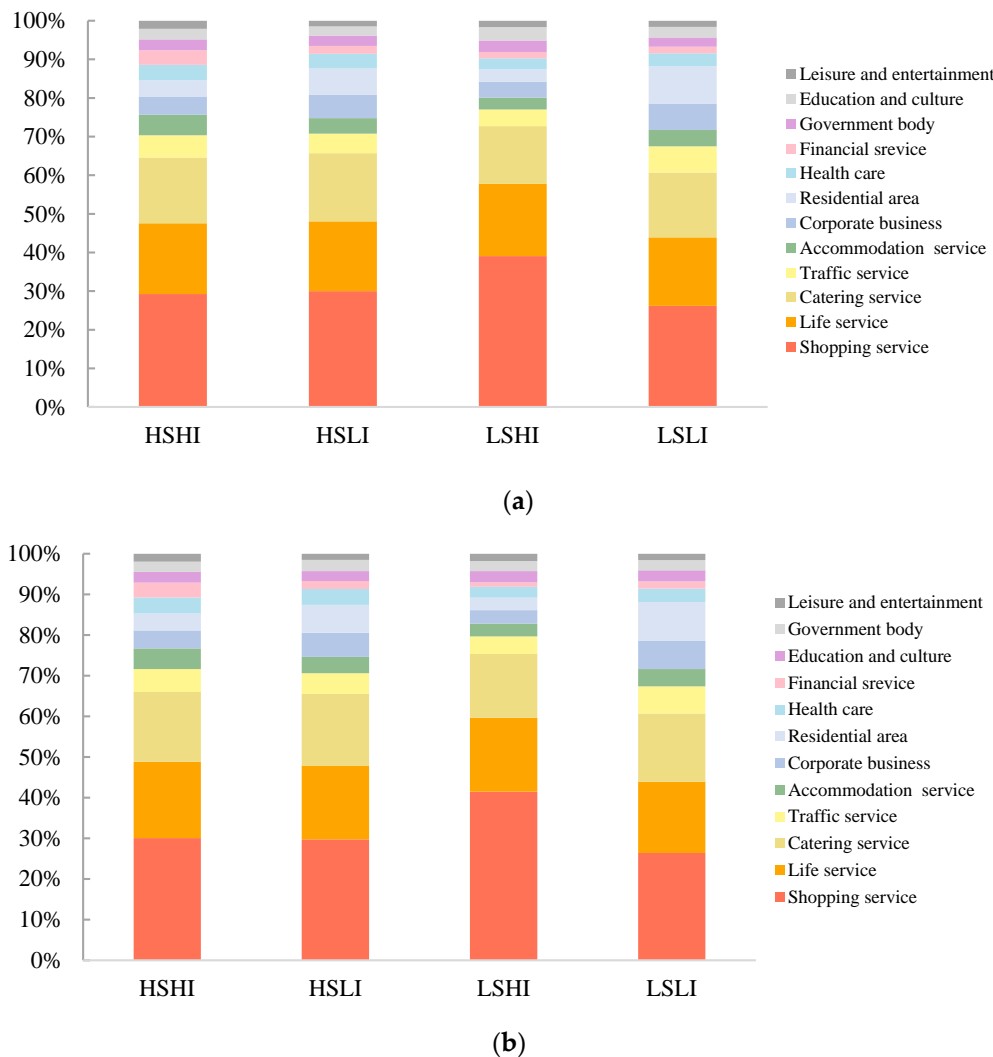

**Figure 11.** Land use composition of street vitality types. (**a**) Weekdays; (**b**) Weekends.

Given that the last eight or nine land use types of each street vitality type account for less than 30% of the cumulative total (see Figure 11), we considered the land use types with a cumulative percentage of 70% as the main land use type. The results show that shopping, life, and catering services are fundamental land use, and the traffic services of the high-vitality streets are prominent. In addition to basic services, low-vitality streets comprise a larger proportion of residential areas. In addition to the main land use types, the proportions of other land use types on HSHI are relatively similar. HSHI creates vitality through the main land use types and stabilizes it through the uniform and coordinated effect of the other land use types.

## 5. Discussion

Most studies have recognized street vitality from a static perspective, defining it as the ability to support various activities, and use the accumulated or average street activities to measure vitality [48,49]. However, in this study, we defined street vitality as the ability to attract and maintain activities at all times. Thus, we focused on the temporal dynamics of street vitality and scientifically evaluated street dynamic vitality from a full perspective. Theoretically, we think that the street dynamic vitality is an ability to attract people and activities while maintaining the attraction at all times, which enriches the previous definition of street vitality and promotes the transformation of street vitality research from static vitality to dynamic vitality. Methodically, we put forward an

evaluation index "vitality stability", and the experimental results verified the necessity and scientific rigor of this indicator. The vitality intensity and vitality stability helped us to fully evaluate the street dynamic vitality. Practically, our results reflect the relationship between street dynamic vitality types and land use, which can offer significant guidance for improving street vitality intensity and stability. We suggest three effective measures to improve street vitality intensity: (1) enhance the intensity of land development and increase the utilization rate of land resources, (2) provide a good consumption environment by building large shopping malls and commercial pedestrian areas as vitality attraction poles, and (3) perfect transportation networks (long-term goal), increase the number of bus routes, and focus on planning and building parking lots (short-term goal), creating convenient travel traffic conditions. To ensure street vitality stability, it is necessary to increase land use types and distribute land use types evenly, which would meet the needs of different activities at different times and enrich shopping, dining, transportation, leisure, and entertainment functions on residential streets, thus creating regional vitality micro-circulation, and minimize the separation of working places and residences to strengthen the stability of street vitality during weekdays.

Despite the many strengths of this study, there are some limitations that future research could address. First, we distributed the population according to the proportion of the spatial area, which resulted in a gap between the estimated and actual number of people. Given the impacts of building properties on population distribution, future research could consider the building properties within the Thiessen polygon to assign weights and more accurately simulate the population at the street level. Second, because human activities show obvious volatility and periodicity [50], the mobile communication dataset which covered four weekdays and two weekends was adequate to reflect activity patterns of an activity cycle, and the result was consistent with the urban spatial structure of the main urban area of Xining, so it was representative to study street vitality. Future works could consider quantifying street dynamic vitality comprehensively by adaptively weighting mobile communication datasets and social media check-in data. Thirdly, we discussed the impact of traffic services on street dynamic vitality, but ignored the particularity of pedestrian streets with high-intensity vitality but fewer traffic POIs. Differentiating street types may help us to unearth the causes of their dynamic vitality of different street types. Finally, although many factors affect street dynamic vitality, our research only focused on the impact of land use on the street dynamic vitality, but other built environmental factors, such as the number of intersections, road network density, distance to traffic facilities, slow traffic environment, and other traffic data, affect the street vitality by affecting traffic accessibility [51,52]. Follow-up research could thus use the traffic accessibility and characteristics of the street fabrics and physiognomy provided by aerial images or light detection and ranging (LiDAR) data to discuss the driving forces behind street dynamic vitality in depth.

## 6. Conclusions

By collecting mobile communication data, we quantified street dynamic vitality in Xining city's main urban area, used vitality intensity and stability to evaluate street dynamic vitality, and explored its driving factors from a land use perspective.

The results fund that the temporal characteristics of street vitality were consistent with human activities: in the main urban area of Xining city, the daytime street vitality was higher than that of the night-time, the daytime street vitality was lower on weekends than on weekdays, and night-time street vitality was higher on weekends. The spatial distribution of street vitality followed the attenuation principle and was centered on large shopping malls. In Xining city, the high vitality streets were mainly concentrated around the shopping malls, commercial pedestrian areas, schools, and rivers. The vitality intensity and stability were important indicators of dynamic vitality; the introduction of vitality stability indicators effectively distinguished streets with similar overall vitality values but different change patterns. We found that the vitality of most streets in the main

urban area of Xining city was characterized by low intensity and low stability, and that vitality intensity on weekdays and weekends was similar; however, street vitality stability increased significantly on weekends. While previous studies have shown that the spatial concentration and orderliness of land use have a significant impact on vitality [53], we found that land use concentration was positively related to street vitality intensity, and its richness and uniformity were positively related to street vitality stability. Superior shopping, life, catering, and transportation services appeared on these high-intensity vitality streets, which confirm the previous findings that convenient traffic conditions are an important driving force for vitality [11]. The streets with low vitality stability and low vitality intensity in the main urban area of Xining city are widely distributed, and it is necessary to improve the street vitality intensity and stability. In addition to the existing large shopping malls, large shopping malls should be built on the fringe of the main urban area, such as in the university town in Chengbei District. As Xining city is a mountainous city, there are limited land resources, and it is difficult to increase the density of its road network. Therefore, the city must make full use of land resources, enrich the types of industries, distribute urban function points more evenly, and improve traffic services by adding more bus routes and parking lots.

**Author Contributions:** Xin Guo and Hongfei Chen conceived and designed the presented idea; Xin Guo implemented the experiments, analyzed the results, and wrote the manuscript; Hongfei Chen and Xiping Yang collected the research data, reviewed the manuscript, and provided comments. All authors have read and agreed to the published version of the manuscript.

**Funding:** This research was jointly funded by MOE (Ministry of Education in China) Project of Humanities Social Sciences (No. 16YJCZH005), Fundamental Research Funds for the Central Universities (No. GK201803052), National Natural Science Foundation of China (No. 41801373), and Open Research Fund Program of LIESMARS (Grant No. 20S03).

**Data Availability Statement:** The data presented in this study are available on request from the corresponding author. The data are not publicly available due to restrictions of privacy and morality.

**Acknowledgments:** The authors thank all the reviewers for their works.

**Conflicts of Interest:** The authors declare no conflict of interest.

## Appendix A

**Table A1.** POI data classification.

| Integration Category | Specific Type | Counts |
|---|---|---|
| Catering service (CS) | Catering services | 8988 |
| Shopping service (SR) | Shopping services/Cars or motorcycle sales/Access facilities/Place name and address information | 15,031 |
| Life service (LS) | Life services/Cars or motorcycle services or repairs/Public facilities/Access facilities/Indoor facilities/Place name and address information | 9406 |
| Traffic service (TS) | Transportation facility services/Access facilities/Place name and address information | 3007 |
| Corporate business (CB) | Access facilities/Place name and address information | 3171 |
| Government body (GB) | Government agencies/Social organizations/Access facilities/Place name and address information | 1371 |
| Education and culture (EC) | Education and culture services/Access facilities/Place name and address information | 1347 |

**Table A1.** *Cont.*

| Integration Category | Specific Type | Counts |
|---|---|---|
| Accommodation service (AS) | Accommodation services/Access facilities | 2219 |
| Residential area (RA) | Business residence/Access facilities | 3878 |
| Health care (HC) | Health care services/Access facilities | 1903 |
| Leisure and entertainment (LE) | Sports and leisure services/Famous tourist sites/Access facilities/Place name and address information | 837 |
| Financial service (FS) | Financial insurance services/Access facilities | 1109 |

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
