# Peer review of "An Evaluation of Street Dynamic Vitality and Its Influential Factors Based on Multi-Source Big Data"

_ijgi, doi:10.3390/ijgi10030143_

Round 1

Reviewer 1 Report

After reading the manuscript An Evaluation of Street Dynamic Vitality and Its Influencing Factors Based on Multi-source Big Data, I highlight next remarks:

  • Abstract. The research aim is unclear, why vitality and land use diversity were examined in the selected city? Please provide the purpose to carry out the study.
  • Introduction. The characterization of the term “urban vitality” presented is very ambiguous. Furthermore, the main factors of “urban vitality” should be determined and properly associated. Shortcomings of previous studies focused on static vitality should be revealed to define the background that bases the development of the study. An in-depth literature review than that displayed in section 2 should help authors to identify gaps in knowledge to be bridged by multi-source big data. Sections 1 to 3 should be accordingly merged to shape a solid introduction.
  • Data and methods. The main urban area of Xining considered in the study should be specified. Besides, criteria to limit the analysis to several days in the month of August which even not complete a whole week should be also provided. Why does such a short time is representative to study street vitality of a city? It is also unclear if data cover 24 hours per day. How can the consideration of one-third of all mobile phone users in the city influence the study? If the spatial sampling accuracy of mobile communication is between 50 and 300 meters, how does this point impact the fact that the 652 streets deemed in the main urban area are less than 200 meters? Despite the built environment conditions significantly affect street vitality, they were ignored in the study. Notations of equations are inconsistent, e.g. Pi (equation 2 and 8), please use different notations for each variable. Most indicators were provided without any prior explanation and correlation to the research. For instance, what is stability (STB)? maybe it refers to instability as “the characteristic denoting how activities maintain balance with time”? Criteria to classify the POI data into 12 categories are very vague, it should be further developed. On the whole, a general overview of the methodology applied is required by linking adopted analysis/indicators. Figure 2 does not allow to perceive the location of base stations on the urban fabric (main streets) of Xining.
  • Results. Figures 3 and 4 are quite ineligible and lines 235 to 255 are overly vague. Thresholds to define high stability and high intensity (HSHI), high stability and low intensity (HSLI), low stability and high intensity (LSHI) and low stability and low intensity are unknown. In table 2, notations of the two Evaluation Indices should follow those of the equations 4 and 5. Title of figure 7 should be enhanced according to the line 293. Beyond the figures shown in Table 3, practical implications of Land-use indices in Xining were not given.
  • Conclusion and discussion. Three suggestions to improve street vitality intensity were offered without any rationale grounded on results.
  • Miscellaneous comments: English vocabulary is quite limited; some terms are repeated even in the same sentence. Abbreviations should be assigned to each category of Appendix A to be consistent with those of Table 3. Some misspelling mistakes were found.

By collecting mobile communication data in a period of six days, the study quantified street dynamic vitality in Xining city’s through the use of a set of given indicators. However, the relevance of the study is highly questioned due to several limitations summarized above that should be accordingly justified. Furthermore, the theoretical and methodological framework should be also strengthened. Beyond the tables and figures displayed, the contribution in the field is scarce because significant implications of results obtained in the city were not disclosed.

Author Response

Thanks very much for your comments. We have fully absorbed your suggestions and tried to solve your doubts. Please review the following document for the response.

Reviewer 2 Report

An Evaluation of Street Dynamic Vitality and Its 2 Influencing Factors Based on Multi-source Big Data

This article analyzes urban vitality, such an important indicator (and predictor) for measuring urban development capacity. It does it by using streets as basic unit of analysis. For this, the authors examined this vitality and the land-use diversity of Xining City’s main urban zone, using mobile communication and points of interest data.

The results obtained in this article are quite obvious: “Results revealed that the temporal change characteristics of street vitality are consistent with human activities: daytime vitality is higher than night vitality, daytime street vitality is lower on weekends than on weekdays, and night street vitality is higher on weekends. The spatial distribution of street vitality follows the attenuation principle and is centered on large shopping malls and squares. As vitality intensity and stability are important indicators of dynamic vitality, the introduction of vitality stability indicators effectively distinguished streets with similar overall vitality values, but different change patterns. By matching different vitality intensity and stability levels, we found that most streets have low intensity and stability characteristics, and a small number can maintain sustainable and relatively high vitality”.

However, the methodology behind the article (by using big data for extracting patterns) presents a great interest and my judgment is favorable to accept this publication after some few suggestions:

  1. The case study is focused on the Chinese Xining City’s main urban zone. The authors introduce the Chinese context in a very brief way at the introduction: “Guided by the concept of people-oriented and harmonious living, Chinese urbanization is transforming from scale development to connotative growth[1]”. I completely recommend extending this sentence by explaining better the “Chinese urbanization process”. For people abroad China, sometimes it is difficult to understand the magnitude of this process. I strongly recommend doing it by using a global context by using global indicators such the shifting centers of gravity of world's human dynamics.
  2. The authors use the streets such the most basic urban public spaces and “their dynamic vitality directly affects urban vitality”. Something that the authors should refer is the relation between street and urban fabrics/physiognomy. I mean, it is not the same a street located in a Central Business District (CBD) of any USA city than a street located in any residential district. For this reason, street characteristics should be related to the urban physiognomy. In this sense, I strongly recommend you check some studies that related the use of street based metrics to characterize urban typologies. It can be good for you to add some reference to this.
  3. The authors “examined this vitality and the land-use diversity of Xining City’s main urban zone, using mobile communication and points of interest data”. Closely related with the last point, it could interesting to correlate your results with the land-use of the city estimated based on physical physiognomy of the city. About the most important methodologies, there are some relevant studies focused on land-use mapping in cities using aerial images and LiDAR data.

Author Response

Thanks very much for your comments. We have absorbed your suggestions and tried to solve your doubts. Please review the following document for the response. 

Reviewer 3 Report

Thank you for providing the opportunities to review your work.

  • Introduction: please provide the definition of Street Dynamic Vitality in this research. You have summarized several studies to bring dynamic vitality and street vitality, but I don't see a clear definition about this term.
  • Related work: once you have a definition, you should address research gap. It seems you tried to distinguish static and dynamic vitality, but how about the methodology? how those have been measured and what are the differences among the studies?
  • Data and Methods: you used somewhat personal records. How the datasets were processed? How did you deal with privacy concerns? What are the protocol such as Internal Review Board?
  • Results: please consider to recreate maps that do not use acronym. The readers might get confused.

Author Response

(The authors gave the same response as above.)

Reviewer 4 Report

The present manuscript is focused on the evaluation of the street dynamic vitality. To this end, the Authors presented a case study in which they have examined the street vitality and the land-use diversity of Xining City’s main urban zone, using mobile communication and points of interest data.
The paper is well organized and the methodology is described in a satisfactory way, although some aspects should be clarified (as subsequently indicated). Some literature references are missing. Finally, the Discussion section could be enriched by addressing the open questions suggested to the Authors.
On the whole, the article can be accepted after a minor revision, according the following comments and suggestions:

Row 114-115: please, reformulate the statement in a clearer way.

Figure 1: I suggest to use "streets" or "road network" instead of "roads" in the Figure legend.

At the beginning of Section 4.1 are listed the data sources used. In this context, I guess that other type of data, such as traffic information, could be useful in order to describe the street vitality: did the Authors considered the possibility to use other additional data sources. Please, add a comment about such possibility.
Also in Section 4.1, considering communication records, obviously we assume that those data have been suitably anonymized, as the Authors have opportunely reported in Table 1. Nevertheless, I suggest to anticipate this aspect and to specify it just in this Section.

Figure 2: it could be useful to rearrange the Figure by splitting it into four different parts, in order to show and depict some examples of the four different data typologies exploited.

Row 142: add a reference for Baidu Map.

Row 148: add a reference for Aude Navigation Open Interface.

Row 158: Thiessen instead of Tyson (please correct hereafter).

In Section 4.2 several formulae are reported. If those formulae come from previous studies, the Authors should add the opportune references. Otherwise, formulae should be introduced with a statement such as "we have defined and proposed the following formulae" (and possibly explained).

Row 197: Add the literature reference for the Xi vitality value calculation or explain how it was calculated.

Row 202: add the literature references for the indexes used (density, richness, Simpson, and main land-use types).

At the beginning of Section 5.1 several city’s districts are listed. It could be useful a Figure depicting the location of the various city Districts (e.g., as Appendix A2).

Section 5.4, Row 329: please, add the reference to the land-use types considered and listed in Appendix A1. The same in Figure 10.

Finally, some additional open questions that I would address to the Authors:
- Concerning the OSM data about road network, is it possible (and effective), for the purposes of your study, to distinguish the ordinary streets from those reserved only to pedestrians?
- As anticipated in a previous comment, I would ask to the Authors if traffic data can be considered as driving factors affecting the street vitality?

Author Response

(The authors gave the same response as above.)

Round 2

Reviewer 1 Report

Most theoretical and methodological flaws summarized in the previous study were not addressed yet. The manuscript is highly confusing and its replication is thus very unlikely. The short period under study highly questions the relevance of the research. Practical implications and contributions in the field are unclear. Authors are suggested to reformulate the investigation by significantly increasing data examined and defining a concrete goal that serves to properly organize the analysis. 

Author Response

Thanks again for your comments! We have revised the manuscript and replied to your comments point-by-point. Please see the file for our response.

Reviewer 3 Report

Thank you for the responses.

I see a lot of improvement and integration of the introduction chapter. This reads better than before.

Please improve the legibility of maps (resolution, size of the map, and legend). It is still difficult to read.

Author Response

Thanks again for your comments! We have revised the manuscript and replied to your comments point-by-point. Please review the file for our response.
